# Engineering of *Saccharomyces cerevisiae* for 24-Methylene-Cholesterol Production

**DOI:** 10.3390/biom11111710

**Published:** 2021-11-17

**Authors:** Jiao Yang, Changfu Li, Yansheng Zhang

**Affiliations:** 1School of Environmental and Chemical Engineering, Shanghai University, Shanghai 200444, China; luckjiaoyang@shu.edu.cn; 2School of Life Sciences, Shanghai University, Shanghai 200444, China; changfuli@shu.edu.cn

**Keywords:** 24-methylene-cholesterol, campesterol, 7-dehydrocholesterol reductase

## Abstract

24-Methylene-cholesterol is a necessary substrate for the biosynthesis of physalin and withanolide, which show promising anticancer activities. It is difficult and costly to prepare 24-methylene-cholesterol via total chemical synthesis. In this study, we engineered the biosynthesis of 24-methylene-cholesterol in *Saccharomyces cerevisiae* by disrupting the two enzymes (i.e., ERG4 and ERG5) in the yeast’s native ergosterol pathway, with ERG5 being replaced with the DHCR7 (7-dehydrocholesterol reductase) enzyme. Three versions of DHCR7 originating from different organisms—including the DHCR7 from *Physalis angulata* (PhDHCR7) newly discovered in this study, as well as the previously reported OsDHCR7 from *Oryza sativa* and XlDHCR7 from *Xenopus laevis*—were assessed for their ability to produce 24-methylene-cholesterol. XlDHCR7 showed the best performance, producing 178 mg/L of 24-methylene-cholesterol via flask-shake cultivation. The yield could be increased up to 225 mg/L, when one additional copy of the *XlDHCR7* expression cassette was integrated into the yeast genome. The 24-methylene-cholesterol-producing strain obtained in this study could serve as a platform for characterizing the downstream enzymes involved in the biosynthesis of physalin or withanolide, given that 24-methylene-cholesterol is a common precursor of these chemicals.

## 1. Introduction

Phytosterols are an important class of plant secondary metabolites, with great potential in the healthcare and food industries. The predominant sterols in plants are β-sitosterol, stigmasterol, and campesterol [1]. As dietary components, phytosterols have been shown to lower cholesterol levels, and have antioxidant effects; they are also widely used as additives in foods and cosmetics [2]. Phytosterols also serve as critical pharmaceutical precursors of hydrocortisone, progesterone, and pregnenolone [3]. 24-Methylene-cholesterol was first isolated from honeybees by Barbier, and was later detected in the pollen tubes of various land plants [4]. Subsequently, some marine organisms—such as mussels and brown seaweeds—were also reported to contain 24-methylene-cholesterol [5]. Recent studies have demonstrated that 24-methylene-cholesterol has moderate antioxidant activity [6]. Importantly, it is a necessary precursor of withanolide and physalins, which have been proven to have the properties of fighting against cancers, inhibiting neurodegenerative diseases, and triggering apoptosis [7,8,9,10].

Undoubtedly, 24-methylene-cholesterol is an important substance in the pharmaceutical industry. However, both direct extraction from plants and chemical synthesis are highly costly and inefficient, in addition to the environmental pollution caused by these approaches. On the other hand, great progress has been made in the production of industrially relevant substances in microbes, allowing for the environmentally friendly, low-cost production of high-purity products. Microbial cell factories provide great opportunities for large-scale production of complex bioactive or industrially important molecules. Many types of microorganisms have been used as sterol producers—for example, *Yarrowia lipolytica* [2] and *Pichia pastoris* [11,12]. Squalene plays a dominant role as a starting material in sterol biosynthesis, which involves reactions catalyzed by a series of enzymes to produce ergosterol [13]. Ergosta-5,7,24-trienol(5-dehydroepisterol) is an important metabolic intermediate in the ergosterol biosynthetic pathway. The 5-dehydroepisterol is transformed into ergosterol via two enzymes (ERG5 and ERG4) (Figure 1). If ERG5 is replaced with a 7-dehydrocholesterol reductase (DHCR7)—an enzyme that reduces the seventh position of the carbon–carbon double bonds of 5-dehydroepisterol [14]—the ergosterol pathway is redirected for the formation of campesterol, and 24-methylene-cholesterol will be further produced once the replacement is accompanied by the inactivation of ERG4 in the yeast (Figure 1). DHCR7 is a membrane-embedded enzyme that is widespread in plants and animals, but is lacking in *Saccharomyces cerevisiae*. DHCR7s from *Oryza sativa* (*OsDHCR7*) and *Xenopus laevis* (*XlDHCR7*) have been previously characterized, and were utilized to produce campesterol in *Y. lipolytica* [14]. Using a 5 L bioreactor, a maximal yield of 837 mg/L was obtained for campesterol production in *Y. lipolytica* [2,15]; however, this low titer of campesterol cannot satisfy large-scale fermentation production. Three approaches are thus advised for further study: firstly, searching for new potential microbial chassis strains [16]; secondly, rational design of a biosynthetic pathway directing more carbon to the target chemical [17]; thirdly, characterizing more genes encoding critical enzymes in the biosynthetic pathway to find highly efficient enzymes [18,19]—for example, *DHCR7*, which plays an important role in the 24-methylene-cholesterol biosynthetic pathway. 

*Physalis angulata* is an annual gramineous herb belonging to the genus *Physalis* of the Solanaceae family. A diverse array of pharmaceutically active compounds has been characterized from *P. angulata* plants, including physalins and their derivatives, withanolide, terpenoids, and flavonoids. *P. angulata* plants are particularly rich in physalin and withanolide, which are derived from 24-methylene-cholesterol [20]. *DHCR7* is necessary for the biosynthesis of 24-methylene-cholesterol; thus, we surmised that the *P. angulata* species may contain a higher *DHCR7* activity.

In this study, we were especially interested in the characterization of the genes encoding DHCR7 from *P. angulata*, because this enzyme catalyzes a critical step in the biosynthesis of 24-methylene-cholesterol (Figure 1). We first identified the gene *PhDHCR7* in *P. angulata* by mining the third-generation transcriptome sequencing data of *P. angulata*. Next, *PhDHCR7* and two heterologous *DHCR7* genes of *O. sativa* and *X. laevis* were codon-optimized and introduced into *S. cerevisiae* to construct a strain producing 24-methylene-cholesterol, which is poorly synthesized by chemical methods. Additionally, we conducted shake-flask fermentation to assess relationships between quantities of intracellular 24-methylene-cholesterol and glucose, and optical density (OD) of cells, in shake-flask cultivation.

## 2. Materials and Methods

### 2.1. Cloning of the Full-Length Coding Region of the PhDHCR7 Gene

Total RNA was extracted using an EASY Spin Plant RNA Rapid Extraction Kit (Aidlab Biotech, Beijing, China). RNA concentration was determined using a NanoDrop 2000C ultra-microspectrophotometer (Thermo Fisher, Massachusetts, USA). First-strand cDNA was synthesized using a PrimeScript™ RT Reagent Kit with gDNA Eraser (Takara, Beijing, China), in accordance with the manufacturer’s protocol. The *PhDHCR7* gene was identified from *P. angulata* by mining the third-generation transcriptome data in-house. The *PhDHCR7* cDNA was amplified using the primers PhDHCR7-F and PhDHCR7-R (Appendix A), designed according to the full-length cDNA sequence of the *PhDHCR7* gene. 

### 2.2. Strains, Media, and Culture Conditions

All of the yeast strains used in this study are listed in Table 1. *S. cerevisiae* strain YS5 was maintained in the authors’ laboratory. Strains were cultured at 30 °C in YPDA liquid medium (10 g/L yeast extract, 20 g/L peptone, 20 g/L glucose, and 0.04 g/L adenine sulfate). After transformation, the resulting colonies were selected on an appropriate synthetic dropout (SD) medium (20 g/L agar, 6.7 g/L yeast nitrogen base without amino acids, 20 g/L glucose, and appropriate amino acid dropout mix). SD-URA (SD medium lacking uracil) was used to select colonies of YS6, YS7, and YS8 strains, each expressing a functional *URA**3* gene. SD-TRP (SD medium lacking tryptophan) was used to select colonies of YS9, YS10, and YS11 strains harboring a *TRP1* expression cassette. SD-HIS (SD medium lacking histidine) was used to screen colonies of YS12 containing a HIS3 selection marker. 

### 2.3. Strains and Plasmid Manipulation

All primers used in this study are listed in the Appendix A. All heterologous genes introduced into *S. cerevisiae* were codon-optimized for expression in the corresponding yeast hosts. Sequences of codon-optimized genes are listed in the Appendix A. These were obtained via DNA synthesis with GenScript and sequence-verified. The 5’ and 3’ flanking regions of the corresponding genes were amplified from yeast genomic DNA. To construct gene knockout fragments, the flanking region, selection marker ORF, and gene ORF were assembled using overlap-extension PCR, and then fragments were ligated into a T-vector (PMD19T, Takara) and sequenced to examine DNA sequence integrity. Transformation of *S. cerevisiae* was performed using the LiAc/SS carrier DNA/PEG method [21]. Transformant selection was carried out on appropriate amino acid dropout media plates based on the selection markers utilized. A Rapid Yeast Genomic DNA Isolation Kit (Sangon Biotech, Shanghai, China) was used to isolate yeast genomic DNA; PCR and sequencing were performed to verify the transformants. 

### 2.4. Extraction and Quantification of Sterols

For each sampling, yeast cells were harvested from 1 mL of the culture by centrifugation. To quantify the sterol content, 0.1 mL of 0.04 mg/mL cholesterol (Solarbio, Beijing, China) was added to the cell pellets as an internal standard. Harvested cells were re-suspended with 20 mL of KOH–methanol solution (20%, *w*/*v*), and the lid was screwed on tightly. The mixture was incubated at 60 °C for 4 h before adding 5 mL of hexane to the saponification liquid and vortexing until uniformly mixed. The above procedure was repeated three times. The hexane extract was evaporated completely. The residue was dissolved in 50 µL of BSTFA (bis(trimethylsilyl)trifluoroacetamide) and incubated at 70 °C for 60 min. The resulting solution was added to 70 µL of hexane and analyzed via gas chromatography–mass spectrometry (GC–MS).

### 2.5. Gas Chromatography–Mass Spectroscopy (GC–MS) Analysis

GC/MS (SHIMADZU QP2010 Ultra equipped with an RTX-5MS capillary column (30 m × 0.25 mm × 0.25 mm) analyses were used to examine the products. The injector temperature was set at 280 °C with splitless injection. The carrier gas was helium, with a flow rate of 1 mL/min and a pressure of 61.3 kPa. The program began with 90 °C held for 1 min, then increased to 300 °C at 30 °C/min for 25 min. The MS ion source temperature was set to 230 °C. Spectra were recorded from *m*/*z* = 50 to *m*/*z* = 800. Mass spectral fragmentation patterns were compared to reference spectra in the NIST library and literature.

### 2.6. 24-Methylene-Cholesterol Production by Shake-Flask Fermentation

Preserved yeast clones were revitalized on YPDA liquid medium at 30 °C and used for inoculation. The shake-flask cultivation medium was YPDA (10 g/L yeast extract, 20 g/L peptone, 20 g/L glucose, and 0.04 g/L adenine sulfate), supplemented with 12.5 g/L KH_2_PO_4_ and 2.5 g/L MgSO_4_·7H_2_O. Fermentation was completed on a rotary shaker at 30 °C and 220 rpm for 6 days. Five milliliters of the culture was sampled every 12 h and, after each sampling, 5 mL of YPDA (containing 12.5 g/L KH_2_PO_4_ and 2.5 g/L MgSO_4_·7H_2_O) was added to the remaining cultures. Importantly, 5 mL of 40% glucose—instead of YPDA—was added at 60 h and 108 h. The sampled cultures were stored at −20 °C in 1 mL aliquots for further analysis. To examine the glucose content, 1 mL of sample was centrifuged, and 100 μL of the supernatant was used for further analysis. 

### 2.7. RT-qPCR Analysis

Total RNA from the mother strain and the two engineered strains producing 24-methylene-cholesterol was isolated using a Qiagen RNeasy Mini Kit (Qiagen, Germantown, MD, USA), and then assayed according to the manufacturer’s instructions. The total RNA obtained was then used as a template. First-strand cDNA was synthesized using a FastQuant cDNA kit (TIANGEN, Beijing, China) and then used for expression analysis of the *DHCR7* gene in the two 24-methylene-cholesterol-producing strains. The primers used for RT-qPCR are listed in the Appendix A.

### 2.8. Statistical Methods

Statistical analyses were performed using Origin 2016 (Chicago, IL, USA). All experiments were repeated three times. The data in Figure 2, Figure 3, Figure 4, Figure 5 and Figure 6 are shown as the mean ± standard deviation. The data in Figure 3, Figure 4, Figure 5 and Figure 6 were analyzed using Student’s *t*-test. Origin software 2016 (Chicago, IL, USA) was used for graph construction. 

## 3. Results

### 3.1. Cloning, Sequencing, and Alignment Analysis of PhDHCR7

The *PhDHCR7* gene sequence was retrieved from a calyx transcriptome database of *P. angulata* using a similarity search program; its cDNA was cloned using primers designed against the ORF. The amplified PhDHCR7 was sequenced to verify accordance with the sequence from the transcriptome. The length of the putative PhDHCR7 cDNA was 1305 bp, putatively encoding a protein with 434 amino acids and a mass of 49.5 kDa. The PhDHCR7 protein exhibited 77.57% amino acid sequence identity with OsDHCR7, but only 29.29% similarity to XlDHCR7, indicating that DHCR7 has clearly diverged evolutionarily between animals and plants. Sequence alignment of DHCR7 from *X. laevis*, *O. sativa*, and *P. angulata* is depicted in Figure 2. The putative NADPH pocket and cholesterol binding site are marked.

### 3.2. Campesterol Biosynthetic Pathway Was Constructed in S. cerevisiae via Blocking ERG5 and Introducing the Codon-Optimized DHCR7s

In the ergosterol biosynthetic pathway, the last two steps are catalyzed by the yeast’s ERG4 and ERG5 enzymes. Ergosta-5,7,24-trienol(5-dehydroepisterol) is desaturated to ergosta-5,7,22,24(28)-tetraen-3-beta-ol by the ERG5 encoding a sterol C-22 desaturase [22]. ERG4 functions as a sterol-C-24(28) reductase, reducing ergosta-5,7,22,24-tetraen-3-beta-ol to produce ergosterol [23]. We speculated that by introducing DHCR7 at the position of ERG5 in the yeast genome, DHCR7 would convert ergosta-5,7,24-trienol into campesterol, as depicted in Figure 1. We tested three different codon-optimized DHCR7 genes, including PhDHCR7, with pTEF2 and tCYC1 as controls. These were co-introduced with the selection marker URA3 into S. cerevisiae strain YS5 at the ERG5 chromosomal position, generating strains YS6, YS7, and YS8 harboring DHCR7 from Oryza sativa (Os), Physalis angulate (Ph), and Xenopus laevis (Xl), respectively.

GC–MS was used to determine the compounds produced in the culture broth of YS6, YS7, and YS8, as shown in Figure 3A,B. In the strain YS7, we found a peak with characteristic ions *m*/*z* 129, 343, 367, and 382 at 16.913 min, indicating that campesterol was produced, and demonstrating that the DHCR7 gene we cloned from P. angulata was active. Additionally, at the same retention time, the campesterol characteristic ions appeared in strains YS6 and YS8. Campesterol was not produced in the control stain YS5 (Figure 3A); the product at 16.753 min produced by the strain YS5 corresponds to ergosterol. The ergosterol product was not detected in the cultures of YS6, YS7, and YS8. Figure 3C showed that the strain YS8 with the DHCR7 from X. laevis achieved a high titer of 178 mg/L when cultured in a test tube with 3 mL of YPDA. These results confirm that the disruption of ERG5 by the introduction of heterologous DHCR7 has the ability to produce campesterol in yeast. Particularly, PhDHCR7 functions as expected, reducing the C-C double bond of ergosta-5,7-dienol at the seven position.

### 3.3. 24-Methylene-Cholesterol Was Further Produced by Disrupting ERG4

According to a previous work, deletion of ERG4 results in accumulation of the precursor ergosta-5,7,22,24(28)-tetraenol [24]. We demonstrated that ergosta-5,7,24-trienol can be reduced to campesterol by introducing heterologous DHCR7 and blocking ERG5. We therefore reasoned that 24-methylene-cholesterol would be formed once *ERG4* was disrupted. Hence, we attempted to disrupt ERG4 via homologous recombination in the strains YS6, YS7, and YS8, hoping to produce 24-methylene-cholesterol.

ERG4 was disrupted in strains YS6, YS7, and YS8 to generate strains YS9, YS10, and YS11, respectively. GC–MS was an efficient tool to detect the 24-methylene-cholesterol product. As depicted in Figure 4, 24-methylene-cholesterol was clearly detected, with characteristic ions *m*/*z* 129, 296, 341, and 386 at 17.213 min in strains YS9, YS10, and YS11. These results illustrate that we successfully constructed yeast strains capable of producing 24-methylene-cholesterol by disrupting ERG4 in strains YS6, YS7, and YS8. However, the titer of 24-methylene-cholesterol was low, and needed to be raised.

### 3.4. Overproduction of 24-Methylene-Cholesterol by Increasing the Number of XlDHCR7 Copies

Elevating critical enzymes in the biosynthetic pathway has proven to be a simple and convenient approach for increasing yield [25]. We reasoned that increasing the number of XlDHCR7 copies may increase 24-methylene-cholesterol content. Another copy of the XlDHCR7 expression cassette with selection marker HIS3 was integrated upstream of the ERG4 (TRP1) position in the YS11 genome, generating the strain YS12 with two copies of XlDHCR7. Figure 5A shows that the YS12 strain has 1.55-fold more transcripts of *XlDHCR7* compared to the YS11 strain. We compared 24-methylene-cholesterol content between the heterologous expression strains—YS11 with one copy of DHCR7, and YS12 with two copies. The results shown in Figure 5B reveal that the strain YS12 produced a higher titer of 24-methylene-cholesterol compared with the single-copy DHCR7 strain YS11. These results demonstrate that elevating critical enzyme expression is an efficient approach for increasing 24-methylene-cholesterol content in yeast.

### 3.5. Characteristics of the Optimal Strain YS12 in Shake-Flask Fermentation

In order to explore the relationship between 24-methylene-cholesterol accumulation and the growth rate of the optimized strain YS12, we performed a shake-flask fermentation experiment in a 250 mL Erlenmeyer flask containing 100 mL of medium. The constitution of the medium is described in the Materials and Methods section. To achieve repeatability and accuracy, we conducted the experiment three times, and the mean results are shown in Figure 6.

We used glucose as the sole carbon source, periodically adding concentrated glucose solution after the glucose in the medium was depleted, and keeping the medium volume constant after sampling. The production of 24-methylene-cholesterol was closely related to the cell growth rate. Biosynthesis of 24-methylene-cholesterol began with cell growth; when cells entered a strong growth period (24–96 h), 24-methylene-cholesterol was generated in large amounts; during the stationary phase at 96–144 h, almost no product was produced. 24-Methylene-cholesterol gradually accumulated, synchronous with cell growth rate. Eventually, a titer of 225 mg/L of 24-methylene-cholesterol yield was achieved after 144 h of cultivation. Additionally, we observed that the glucose in the medium was consumed quickly. The strain grew quickly, and the glucose concentration of the medium was too low to satisfy cell growth.

## 4. Discussion

This study is the first report on cloning and functional analysis of a *DHCR7* gene (*PhDHCR7*) from *P. angulate*, which is well known to accumulate abundant 24-methylene-cholesterol-derived compounds, such as physalin and withanolide. To the best of our knowledge, *PhDHCR7* is the second *DHCR7* gene isolated from plant species to date, with the first being *OsDHCR7* from *Oryza sativa* [26]. Given that DHCR7 is a critical enzyme in the engineering steps for 24-methylene-cholesterol production (Figure 1), discovery of *PhDHCR7* can provide an additional gene resource for engineering purposes. Successful production of campesterol (Figure 3) or 24-methylene-cholesterol (Figure 4) in the yeast strains expressing the *PhDHCR7* demonstrated that PhDHCR7 could accept the yeast’s native metabolite 5-dehydroepisterol as a substrate (Figure 1). Next, we assessed PhDHCR7 for its efficiency in producing campesterol or 24-methylene-cholesterol in the yeast, in comparison with OsDHCR7 from *O. sativa* and XlDHCR7 from *Xenopus laevis*. In order to minimize the variations in the protein translations probably introduced by the difference in codon usage, the three DHCR7s were all codon-optimized based on their *S. cerevisiae* preference, and their expression cassettes were integrated into the yeast genome using exactly the same approach. Similar levels of campesterol (Figure 3) or 24-methylene-cholesterol (Figure 4) were produced when PhDHCR7 or OsDHCR7 was expressed, suggesting that both enzymes exhibited comparable activities. By contrast, XlDHCR7 led to significantly higher levels of campesterol or 24-methylene-cholesterol, compared to PhDHCR7 or OsDHCR7 (Figure 3 and Figure 4). These data are consistent with a previous report, in which XlDHCR7 produced higher levels of campesterol than OsDHCR7 in a *Yarrowia lipolytica* strain [2]. The higher production of campesterol or 24-methylene-cholesterol by XlDHCR7 suggests that it functions more efficiently than PhDHCR7 or OsDHCR7. Yuan et al. predicted the XlDHCR7 protein structure based on homology modeling, and the residues interacting with sterol acceptors were revealed by the molecular docking approach [2]. Both PhDHCR7 and OsDHCR7 share very similar sterol-acceptor-interacting residues, whereas they are distinct in XlDHCR7; in particular, in the positions of 388–391 (numbering in XlDHCR7), the sterol-interacting residue ‘GDLM’ in XlDHCR7 is replaced with ‘PEIL’ in the equivalent positions of PhDHCR7 or OsDHCR7 (Figure 2). The substitution within the sterol-acceptor-interacting residues might offer a plausible explanation of the difference in activity between XlDHCR7 and PhDHCR7 or OsDHCR7. Indeed, Yuan et al. reported that a D389E mutation significantly decreased the XlDHCR7 activity [2]. 

24-Methylene-cholesterol is an important precursor to many steroidal compounds, such as physalin and withanolide [20], and also displays various pharmacological activities [27]. Without any pathway optimizations, the introduction of XlDHCR7 into the YS5 yeast background could achieve a high yield (225 mg/L) of 24-methylene-cholesterol (Figure 6), most likely due to the use of the YS5 strain, which has a greater isoprenoid carbon flux [21]. To the best of our knowledge, this is the highest yield for 24-methylene-cholesterol reported thus far in microbial engineering. We noted that there is still significant room for improvement of the 24-methylene-cholesterol yield in the YS5 background. For example, the yield can be improved by increasing the precursor flux through boosting the yeast mevalonate pathway, as this strategy has been successful in many cases to increase the biosynthesis of diverse steroids [28]. Alternatively, the yield of 24-methylene-cholesterol can be enhanced by modifying the size of the lipid body in yeast. We presumed that 24-methylene-cholesterol, like campesterol [15], may primarily accumulate in the lipid body because of its hydrophobic properties. Furthermore, the successful construction of a 24-methylene-cholesterol-producing yeast strain in this study could serve as a chassis for further characterizing the downstream enzymes that utilize 24-methylene-cholesterol as a starting point to synthesize more complex steroids (e.g., physalin or withanolide), since the pathway beyond 24-methylene-cholesterol toward the biosynthesis of physalin or withanolide is completely unknown.

## Figures and Tables

**Figure 1 biomolecules-11-01710-f001:**
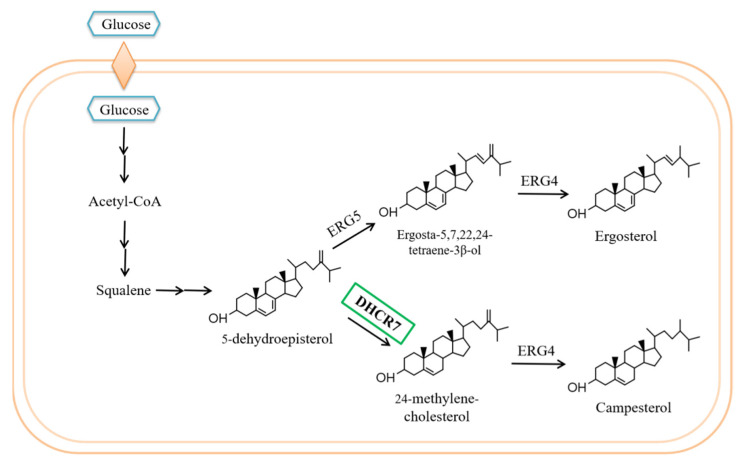
Schematic diagram illustrating the construction of the campesterol and 24-methylene-cholesterol biosynthesis pathways, based on the native ergosterol biosynthesis pathway in *Saccharomyces cerevisiae*. The campesterol biosynthesis pathway was constructed by disrupting ERG5 and expressing the heterologous *7-dehydrocholesterol reductase* gene (*DHCR7*). The 24-methylene-cholesterol biosynthetic pathway was constructed from the campesterol biosynthesis pathway via the disruption of ERG4. Sterol biosynthesis uses a common acetyl-CoA precursor.

**Figure 2 biomolecules-11-01710-f002:**
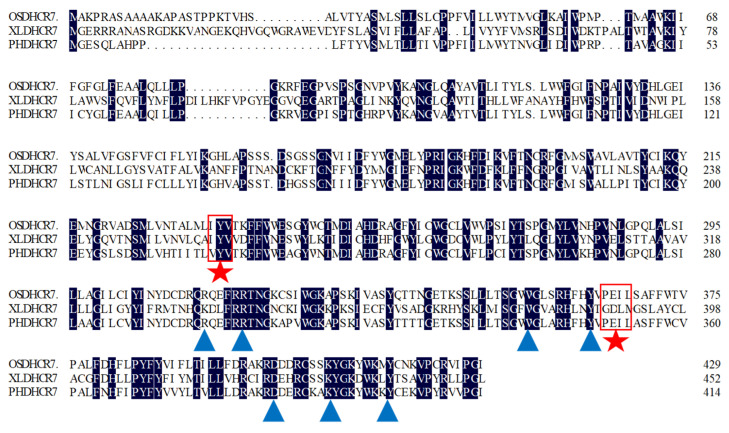
Full-length amino acid sequence alignment of DHCR7s from *Physalis angulata*, *Xenopus laevis*, and *Oryza sativa*. Blue triangles represent the putative NADPH binding sites; red pentagrams represent the putative binding sites for the hydroxyl groups of sterol acceptors.

**Figure 3 biomolecules-11-01710-f003:**
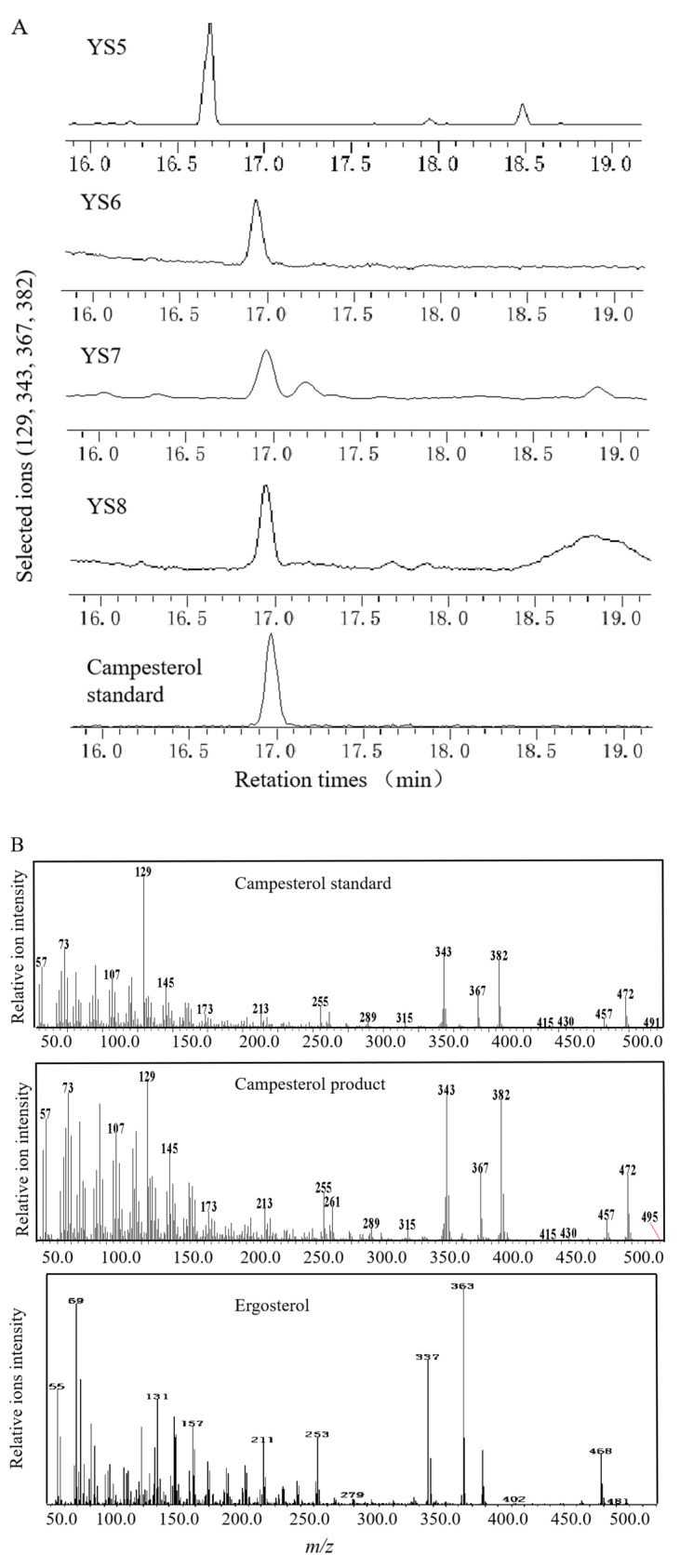
Identification of fermentation products in the recombinant strains via gas chromatography–mass spectrometry (GC–MS): (**A**) GC–MS-extracted ion profiles of the control strain YS5 producing ergosterol and YS6–8 strains producing the campesterol product. (**B**) Mass-fragmented patterns of the campesterol and ergosterol products. (**C**) Quantification of the campesterol product extracted from the strains of YS6, YS7, and YS8. Error bars represent standard deviations (*n* = 3). Asterisks indicate significant differences compared to YS6 and YS7; Student’s *t*-test, ** p* < 0.05.

**Figure 4 biomolecules-11-01710-f004:**
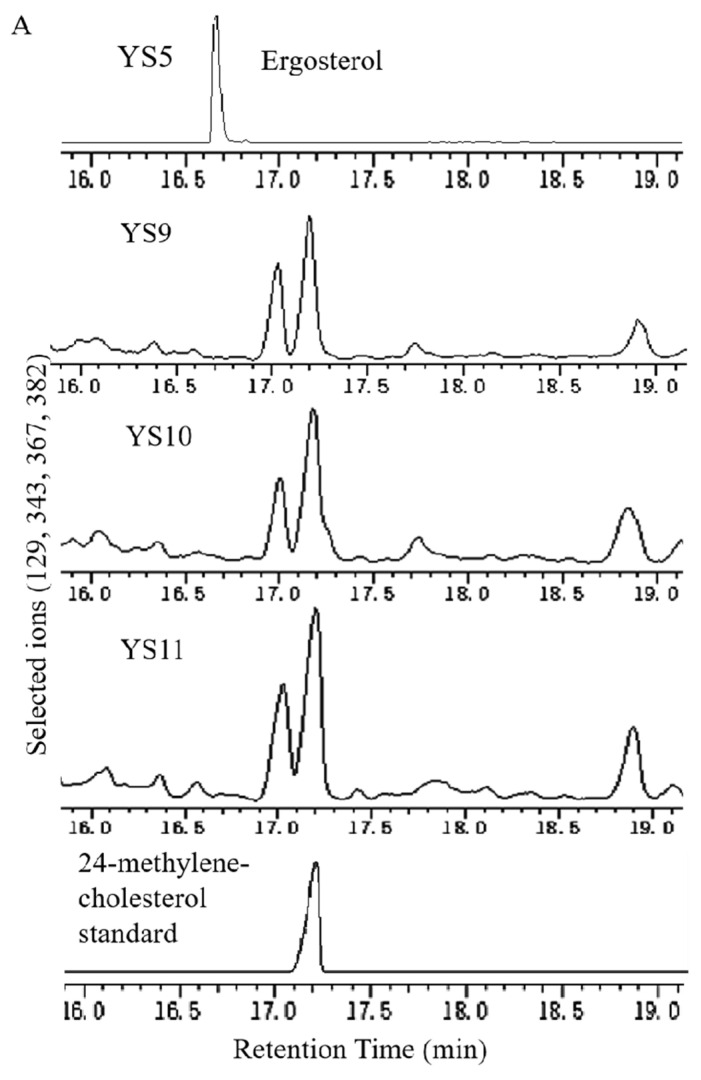
Identification of fermentation products in recombinant yeast strains via gas chromatography–mass spectrometry (GC–MS): (**A**) GC–MS patterns of the parent strain YS5 producing ergosterol, and YS9–11 strains producing the 24-methylene-cholesterol product. (**B**) Mass chromatography of the 24-methylene-cholesterol product as well as its authentic standard. (**C**) Quantification of the 24-methylene-cholesterol produced by the strains YS9, YS10, and YS11. Error bars represent standard deviations (*n* = 3). Asterisks indicate significant differences compared to YS9 and YS10; Student’s *t*-test, ** p* < 0.05.

**Figure 5 biomolecules-11-01710-f005:**
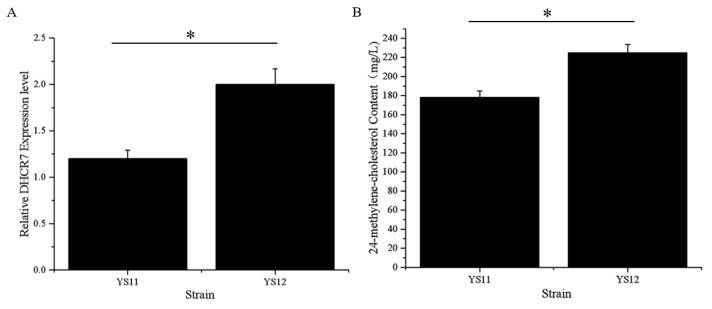
Real-time PCR analysis of XlDHCR7 in strains YS11 and YS12, with different 24-methylene-cholesterol yields: (**A**) YS12 has 1.55-fold higher mRNA levels of XlDHCR7 compared to YS11. (**B**) 24-Methylene-cholesterol content in the strains with heterologous expression of XlDHCR7—YS12 compared with YS11. An additional copy of XlDHCR7 increased 24-methylene-cholesterol production by 23%. Error bars represent standard deviations (n = 3). Asterisks indicate significant differences compared to YS11; Student’s *t*-test, ** p* < 0.05.

**Figure 6 biomolecules-11-01710-f006:**
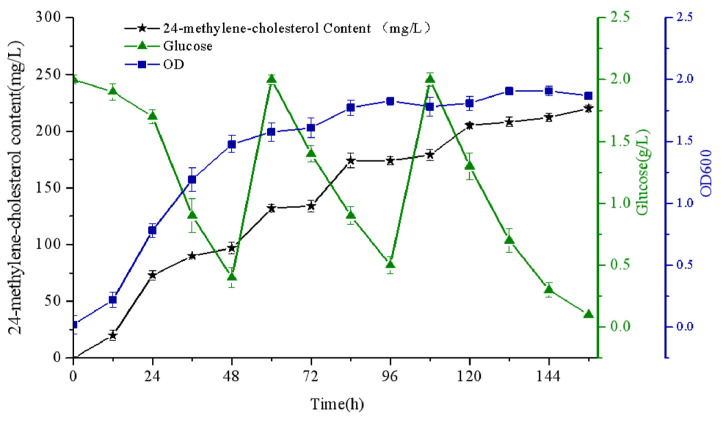
Characteristics of the optimal strain YS12 in shake-flask fermentation with glucose. Error bars represent standard deviations (*n* = 3).

**Table 1 biomolecules-11-01710-t001:** Strains and plasmids used in this study.

Strain Genotypes and Corresponding Products in This Study			
Strains	Source	Genotype	Major sterol
YS5	[21]	MATα(leu2-3,112 trp1-1 can1-100 ura3-1 ade2-1 his3-11,15)	Ergosterol
YS6	This study	ERG5::URA3-pTEF2-DHCR7(*Oryza sativa*)-tCYC1	Campesterol
YS7	This study	ERG5::URA3-pTEF2-DHCR7 (*Physalis angulate*)-tCYC1	Campesterol
YS8	This study	ERG5::URA3-pTEF2-DHCR7(*Xenopus laevis*)-tCYC1	Campesterol
YS9	This study	ERG5::URA3-pTEF2-DHCR7(*Oryza sativa*)-tCYC1 ERG4::TRP1	24-Methylene-cholesterol
YS10	This study	ERG5::URA3-pTEF2-DHCR7(*Physalis angulate*)-tCYC1 ERG4::TRP1	24-Methylene-cholesterol
YS11	This study	ERG5::URA3-pTEF2-DHCR7(*Xenopus laevis*)-tCYC1 ERG4::TRP1	24-Methylene-cholesterol
YS12	This study	ERG5::URA3-pTEF2-DHCR7(*Xenopus laevis*)-tCYC1 ERG4::TRP1ERG4::HIS3-pTEF2-DHCR7(*Xenopus laevis*)-tCYC1	24-Methylene-cholesterol

## Data Availability

The data presented in this study are available in the main text, figures, tables and Appendix A.

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
