# Peer review of "Engineering of Saccharomyces cerevisiae for 24-Methylene-Cholesterol Production"

_biomolecules, 2021, doi:10.3390/biom11111710_

Round 1

Reviewer 1 Report

Please revise the paper according to the detailed comments below.

1.

The composition of YPDA medium is not same in two sentences. Please check and revise.

-Page 3 of 13 (2.2)

Strains were cultured at 30℃ in YPDA liquid medium (10 g/L yeast extract, 20 g/L peptone and 20g/L glucose).

-Page 4 of 13 (3.6)

The shake-flask cultivation media was YPDA containing 40 g/L glucose, 6.7 g/L yeast nitrogen base (without amino acids or ammonium sulfate), 2 g/L yeast extract powder, 12.5 g/L KH2PO4 and 2.5 g/L MgSO4•7H2O.

2.

In Figure 6, the initial glucose concentration (t=0) is about 2.0 g/L. In addition, another two highest glucose concentration during fermentation are also about 2.0 g/L (t=60, t=108). In the page 4 of 13 (3.6), authors described YPDA liquid medium containing 40 g/L glucose, which was used in the fermentation. If authors used YPDA liquid medium, is it reasonable that the initial glucose concentration is 40 g/L?     

3.

Page 9 of 13 (3.5)

Please describe the concentration of concentrated glucose concentration? In addition, describe the volume of concentration glucose solution fed during fermentation.

4.

Please add the space between digit and unit.

Page 1 of 13 (Abstract), 225mg/L

Page 3 of 13 (2.2)

10g/L yeast extract, 20g/L peptone and 20g/L glucose

20g/L agar, 1.7g/L yeast nitrogen base without amino acids, 5g/L ammonium sulfate, 20g/L glucose and appropriate amino acid dropout mix

Page 9 of 13 (3.5) 225mg/L

Author Response

Dear Reviewers,

Thanks very much for taking your time to review my manuscript. I sincerely appreciate all your comments and suggestions! Attachment is my manuscript revised version. 

Question 1.

The composition of YPDA medium is not same in two sentences. Please check and revise.

-Page 3 of 13 (2.2)

Strains were cultured at 30℃ in YPDA liquid medium (10 g/L yeast extract, 20 g/L peptone and 20g/L glucose).

-Page 4 of 13 (3.6)

The shake-flask cultivation media was YPDA containing 40 g/L glucose, 6.7 g/L yeast nitrogen base (without amino acids or ammonium sulfate), 2 g/L yeast extract powder, 12.5 g/L KH2PO4 and 2.5 g/L MgSO4•7H2O.

Response 1 : Thanks for your comments. we have corrected the sentences as you suggested.

-Page 3 of 13 (2.2)

Strains were cultured at 30 â„ƒ in YPDA liquid medium (10 g/L yeast extract, 20 g/L peptone, 20 g/L glucose and o.o4 g/L adenine sulfate).

-Page 4 of 13 (3.6)

The shake-flask cultivation media was YPDA (10 g/L yeast extract, 20 g/L peptone, 20 g/L glucose and o.o4 g/L adenine sulfate), additionally, added 12.5 g/L KH2PO4 and 2.5 g/L MgSO4•7H2O.

Question 2.

In Figure 6, the initial glucose concentration (t=0) is about 2.0 g/L. In addition, another two highest glucose concentration during fermentation are also about 2.0 g/L (t=60, t=108). In the page 4 of 13 (3.6), authors described YPDA liquid medium containing 40 g/L glucose, which was used in the fermentation. If authors used YPDA liquid medium, is it reasonable that the initial glucose concentration is 40 g/L?

Response 2: Sorry for the mistake, we have corrected it in the revised version. During the fermentation, the glucose was consumed by the yeast strains, and after the 60 hours and 108 hours of the yeast growth, the glucose concentration was decreased to be only about 2.0 g/L, just because of the glucose comsumption by the cells.

Question 3.

Page 9 of 13 (3.5)

Please describe the concentration of concentrated glucose concentration? In addition, describe the volume of concentration glucose solution fed during fermentation.

 Response 3: The concentration of the concentrated glucose is 40%, and after 60 and 108 hours of the yeast growing, 5 mL of 40% glucose solution was added into the culture. These details have been added in the revised version.   

Question 4.

Please add the space between digit and unit.

Page 1 of 13 (Abstract), 225mg/L

Page 3 of 13 (2.2)

10g/L yeast extract, 20g/L peptone and 20g/L glucose

20g/L agar, 1.7g/L yeast nitrogen base without amino acids, 5g/L ammonium sulfate, 20g/L glucose and appropriate amino acid dropout mix

Page 9 of 13 (3.5) 225mg/L

Response 4: Thanks for your comments, they were revised as you suggested.

Reviewer 2 Report

The manuscript describes functional expression of DHCR7 from 3 different organisms (Physalis angulata, Oryza sativa and Xenopus laevis) in the yeast Saccharomyces cerevisiae. This heterologous expression together with deletions of two genes encoding two enzymes involved in the synthesis of ergosterol (ERG5 and ERG4) led to increased production of 24-methylene-cholesterol, which could be potentially used as a precursor for production of other pharmaceutically important steroid compounds. Although the obtained data are interesting, there are several major and minor points that should be improved/provided prior publishing:

Major points:

  1. Results shown in Fig. 3 and 4: It is not clear in which volume where cells cultivated – should be clarified and more precisely described in MaM? Also, the results obtained for a original strain YS5 non-expressing DHCR7 should be shown as a negative control in the analysis.
  2. 5A – shown is a relative expression of DHCR7, but it is not clear the relation to which strain? Should be clarified. The text reffering to Fig. 5A should revised – YS1 strain is not used in the study and expressions much higher should be expressed by particular numbers.
  3. The discussion should be improved overall, there are redundancy of information – paragraph 1 and 2 and the results obtained by authors are, in my opinion, discussed insufficiently. Clearly should be stated what is new in this study and what was obtained previously as concern XlDHCR7 and OsDHCR7.
  4. Authors refer to “the selection marker tryptophan” (first paragraph of discussion), but it is the TRP1 gene which complements trp1-1 mutation.
  5. 2 – the sequence of Rattus norvegicus is missing in the sequence alignment to which authors refer in the text.
  6. Abstract should be improved – authors obtained higher level of 24-methylene-cholesterol production only with XlDHCR7 gene, this should be specified and not in 250 ml media, but in 250 ml flask with 100 ml media.
  7. In Fig. 3C, 4C, 5B, 6 is shown content of campesterol/24-methylene-cholesterol – should be clarified if it corresponds to total content in cells or in media with cells? Comment: 24-methylene-cholesterol is an insoluble compound, how it can be used, when produced in yeast cells and stored in peroxisomes (suggested in discussion), as a precursor for other synthesis?

Minor points:

  1. The text should be revised as in many cases the space between numbers and unites is missing.
  2. Renumber chapters in MaM – should start with number 2 and not 3.
  3. Chapter 2.2 – there are inconsistencies in names of media SD-TRP and SD-URA-TRP (SD medium lacking leucine)???
  4. The correct name is Xenopus and not Xenapus.
  5. Table 1: description of strains is not precise – YS9-12 – the gene TRP should be specified by number TRP1? YS12 – the description of integration is not clear, also in the text.

Author Response

Dear Reviewers,

Thanks very much for taking your time to review my manuscript. I sincerely appreciate all your comments and suggestions! Attachment is my revised manusript.

Major points:

  1. Results shown in Fig. 3 and 4: It is not clear in which volume where cells cultivated – should be clarified and more precisely described in MaM? Also, the results obtained for a original strain YS5 non-expressing DHCR7 should be shown as a negative control in the analysis.

         Response 1: Thanks, all the requested information has been added                       into the section of method and materials. The data from the original                     strain YS5 was included as a negative control.

  1. 5A – shown is a relative expression of DHCR7, but it is not clear the relation to which strain? Should be clarified. The text reffering to Fig. 5A should revised – YS1 strain is not used in the study and expressions much higher should be expressed by particular numbers.

         Response 2: Thanks for your comments, the transcript level of DHCR7 was           compared between the strain YS11 and YS12, and the particular numbers             were added to express the higher levels. All these information has been               included into the revised version.

  1. The discussion should be improved overall, there are redundancy of information – paragraph 1 and 2 and the results obtained by authors are, in my opinion, discussed insufficiently. Clearly should be stated what is new in this study and what was obtained previously as concern XlDHCR7 and OsDHCR7.

     Response 3: Thanks for comments, and we have revised the discussion                 section.

  1. Authors refer to “the selection marker tryptophan” (first paragraph of discussion), but it is the TRP1 gene which complements trp1-1 mutation.

    Response 4: Thanks, we have corrected it.

  1. 2 – the sequence of Rattus norvegicus is missing in the sequence alignment to which authors refer in the text.

         Response 5: Yes, the sequence of Rattus norvegicus was not aligned, and             thus, we corrected the sentence.

  1. Abstract should be improved – authors obtained higher level of 24-methylene-cholesterol production only with XlDHCR7 gene, this should be specified and not in 250 ml media, but in 250 ml flask with 100 ml media.

          Response 6: Thanks, we improved the abstract as you suggested.

  1. In Fig. 3C, 4C, 5B, 6 is shown content of campesterol/24-methylene-cholesterol – should be clarified if it corresponds to total content in cells or in media with cells? Comment: 24-methylene-cholesterol is an insoluble compound, how it can be used, when produced in yeast cells and stored in peroxisomes (suggested in discussion), as a precursor for other synthesis?

         Response 7: thanks, the campesterol/24-methylenecholesterol                             products accumulated within the cells but not in the media, we have                     clarified this by adding more sentences in the section of method and                   materials. As an insoluble compound, the 24-methylenecholesterol                       product may attach to the membranes, such as ER and peroxisome                       membranes, on which it may be oxidized to some end sterol products                   by membrane-associated downstream enzymes, such as cytochrome                     P450s.

Minor points:

  1. The text should be revised as in many cases the space between numbers and unites is missing. 

       Response 1: Thanks for your suggestion. we have checked and revised all             this type of mistakes.

  1. Renumber chapters in MaM – should start with number 2 and not 3.

         Response 2: Sorry for the mistake, we corrected it.

  1. Chapter 2.2 – there are inconsistencies in names of media SD-TRP and SD-URA-TRP (SD medium lacking leucine)???

         Response 3: Thanks for your pointing out these mistakes, and we have                 corrected them in the revised version.

  1. The correct name is Xenopus and not Xenapus.

        Response 4: Thanks, corrected as you suggested.

  1. Table 1: description of strains is not precise – YS9-12 – the gene TRP should be specified by number TRP1? YS12 – the description of integration is not clear, also in the text.

        Response 5: Sorry for missing those information, and they have been                  added in the revised version.

Round 2

Reviewer 2 Report

Although authors improved the manuscript in most of the points I have commented in the first review, there are still several inconsistencies that must be corrected and improved before publishing:

Major comment:

  1. The discussion was improved only partially, especially as concern the logic of information provided. Authors are mixing data that they obtained for PhDHCR7 and XlDHCR7 – e.g. the first paragraph – the titter 225 mg/L was obtained for PhDHCR7 and not for XlDHCR7. The new sentence .......Others increased campesterol accu-mulation by overexpression of dhcr7 in diffenent strains increased campesterol produc-tion by 16 to 77%, the YL-D+D+M-E- was able to achieve the highest campesterol pro-duction with a 1.4-fold improvement over the control YL-D+M-E-, but the YL-D+D+E- strain was found just increased 16% compared with YL-D+E- strain, which is consistent with our results. [15]. Obviously, the multifunctional enzyme (MFE1) which involved in peroxisomal b-oxidation contributed much to increase campesterol productovity......is not understandable – what strains are described there? Abbreviations YL-D+D+M-E etc. are not clear….and  to which particular results it is consistent - it should be stated

Discussion should be written in respect to authors’ results, but there are no references e.g. to figures/numbers that authors obtained, except above titter.

I would recommend to check the English by a native speaker, especially the paragraph concerning peroxisomes.

…. While, the major of 24-methylene-cholesterol stored at peroxisome making it hardly to use as pre-cursors. Numerous researches have demonstrated that peroxisomal localization signal (ePTS1) is efficient to reorient cytoplasmic proteins into the peroxisome[27], additionally, a little 24-methylene-cholesterol diffusing in cytoplasm, it is fully satisfied to characterize enzyme ability

There are particular information, but the logic way how authors are providing this is not clear and understandable, how it is relevant to their work.

Minor points:

  1. Abstract – Spaces are missing between the Latin names and parentheses and 100 ml YPDA

Expression much more must be replaced by particular numbers – the difference is 20 mg/L ( Fig. 4), is it too subjective if it is MUCH more

  1. Why do you use description of genes for names of media – e.g. SD-URA3 ? I guess the medium is without uracil (final product of the uracil synthesis pathway), but strains are complemented with the cassette containing URA3 gene (one of the set of genes encoding enzymes involved in the uracil synthesis). Should not be mixed.
  2. 6 – specify SMALL volume of culture that was used for glucose estimation; last sentence - FOR future analysis – FOR is missing
  3. 2. – …..the product at XX time….. – specify XX time
  4. 4.- title – should be a capital X in XlDHCR7
  5. 4. – replace had for to have in the sentence starting with Figure 5A showed….
  6. 5 – check the titles of y axis (capital letters are not necessary)

Author Response

Major comment:

  1. The discussion was improved only partially, especially as concern the logic of information provided. Authors are mixing data that they obtained for PhDHCR7 and XlDHCR7 – e.g. the first paragraph – the titter 225 mg/L was obtained for PhDHCR7 and not for XlDHCR7. The new sentence .......Others increased campesterol accu-mulation by overexpression of dhcr7 in diffenent strains increased campesterol produc-tion by 16 to 77%, the YL-D+D+M-E- was able to achieve the highest campesterol pro-duction with a 1.4-fold improvement over the control YL-D+M-E-, but the YL-D+D+E- strain was found just increased 16% compared with YL-D+E- strain, which is consistent with our results. [15]. Obviously, the multifunctional enzyme (MFE1) which involved in peroxisomal b-oxidation contributed much to increase campesterol productovity......is not understandable – what strains are described there? Abbreviations YL-D+D+M-E etc. are not clear….and  to which particular results it is consistent - it should be stated

 Answer: Sorry for the confusion, we rewrote the discussion.

Discussion should be written in respect to authors’ results, but there are no references e.g. to figures/numbers that authors obtained, except above titter.

Answer: Thanks for your comment, we rewrote the discussion. 

I would recommend to check the English by a native speaker, especially the paragraph concerning peroxisomes.

…. While, the major of 24-methylene-cholesterol stored at peroxisome making it hardly to use as pre-cursors. Numerous researches have demonstrated that peroxisomal localization signal (ePTS1) is efficient to reorient cytoplasmic proteins into the peroxisome[27], additionally, a little 24-methylene-cholesterol diffusing in cytoplasm, it is fully satisfied to characterize enzyme ability

There are particular information, but the logic way how authors are providing this is not clear and understandable, how it is relevant to their work.

Answer: We agreed with you, and rewrote the discussion.

Minor points:

  1. Abstract – Spaces are missing between the Latin names and parentheses and 100 ml YPDAExpression much more must be replaced by particular numbers – the difference is 20 mg/L ( Fig. 4), is it too subjective if it is MUCH more

        Response 1 : Thanks. we rewrote the abstract .

  1. Why do you use description of genes for names of media – e.g. SD-URA3 ? I guess the medium is without uracil (final product of the uracil synthesis pathway), but strains are complemented with the cassette containing URA3 gene (one of the set of genes encoding enzymes involved in the uracil synthesis). Should not be mixed.

         Response 2 : Thanks for your comments, they were corrected as you                     suggested.

  1. 6 – specify SMALL volume of culture that was used for glucose estimation; last sentence - FOR future analysis – FOR is missing

         Response 3 : Thanks, we have corrected it.

  1. – …..the product at XX time….. – specify XX time

      Response 4 : Sorry for the mistake, we have specified it in the revised version.

  1. - title – should be a capital X in XlDHCR7

     Response 5 : Thanks for your suggestion. we have corrected it.

  1. – replace had for to have in the sentence starting with Figure 5A showed….

      Response 6: Thanks for the mistake, we have corrected it.

  1. 5 – check the titles of y axis (capital letters are not necessary)

     Response 7: Thanks, we have corrected it.